# Large-Scale Rice Mutant Establishment and High-Throughput Mutant Manipulation Help Advance Rice Functional Genomics

**DOI:** 10.3390/plants14101492

**Published:** 2025-05-16

**Authors:** Eyob Kassaye Wolella, Zhen Cheng, Mengyuan Li, Dandan Xia, Jianwei Zhang, Liu Duan, Li Liu, Zhiyong Li, Jian Zhang

**Affiliations:** 1State Key Laboratory of Rice Biology and Breeding, China National Rice Research Institute, Hangzhou 311400, China; eukassa2006@gmail.com (E.K.W.); zhenc0311@gmail.com (Z.C.); 2Department of Biology, College of Natural and Computational Sciences, Debre Tabor University, Debre Tabor P.O. Box 272, Ethiopia; 3School of Life Sciences, Hubei University, Wuhan 430062, China; duanliu@hubu.edu.cn (L.D.); liuli2020@hubu.edu.cn (L.L.); 4College of Life Science and Technology, Huazhong Agricultural University, Wuhan 430070, China; limengyuan0209@webmail.hzau.edu.cn (M.L.); dandanxia@webmail.hzau.edu.cn (D.X.); jzhang@mail.hzau.edu.cn (J.Z.); 5National Nanfan Research Institute (Sanya), Chinese Academy of Agricultural Sciences, Sanya 572024, China

**Keywords:** rice (*Oryza sativa* L.), mutagenesis, loss-of-function, gain-of-function, mutant library and DNA barcoding

## Abstract

Rice (*Oryza sativa* L.) is a stable food for over half of the world population, contributing 50–80% of the daily calorie intake. The completion of rice genome sequencing marks a significant milestone in understanding functional genomics, yet the systematic identification of gene functions remains a bottleneck for rice improvement. Large-scale mutant libraries in which the functions of genes are lost or gained (e.g., through chemical/physical treatments, T-DNA, transposons, RNAi, CRISPR/Cas9) have proven to be powerful tools for the systematic linking of genotypes to phenotypes. So far, using different mutagenesis approaches, a million mutant lines have been established and about 5–10% of the predicted rice gene functions have been identified due to the high demands of labor and low-throughput utilization. DNA-barcoding-based large-scale mutagenesis offers unprecedented precision and scalability in functional genomics. This review summarizes large-scale loss-of-function and gain-of-function mutant library development approaches and emphasizes the integration of DNA barcoding for pooled analysis. Unique DNA barcodes can be tagged to transposons/retrotransposons, DNA constructs, miRNA/siRNA, gRNA, and cDNA, allowing for pooling analysis and the assignment of functions to genes that cause phenotype alterations. In addition, the integration of high-throughput phenotyping and OMICS technologies can accelerate the identification of gene functions.

## 1. Introduction

Rice (*Oryza sativa L.*) is a stable food for over half of the world population, contributing 50–80% of the daily calorie intake [1]. As the most widely used model organism among major crops, rice is essential for genetic and physiological research [2] due to its small genome size (420 Mb) and synteny with other plants [3,4]. Traditional breeding struggles to address yields, stress tolerance, and nutritional traits, necessitating advanced mutagenesis. The annual rice production growth has slowed to the point that it is no longer able to keep up with the growing global population of consumers [5].

In the post-genome-sequencing era, the systematic identification of gene functions and understanding of the molecular mechanisms through which genes regulate important traits are the basis for rice genetic improvements [6]. The construction and phenotypic screening of random loss-of-function or gain-of-function mutant libraries have been regarded as one of the most promising ways to identify functional genes and genetically improve rice varieties [7,8].

According to the Food and Agriculture Organization of the United Nations/International Atomic Energy Agency [9], approximately 1351 mutant rice varieties, related to major agronomic and stress tolerance, have been developed using various mutagenesis techniques and have been released for cultivation, which has helped to address the growing challenges of rice production, including the need for improved agronomic traits, nutritional quality, and resilience to environmental stresses. Despite these advancements, the current global release of rice varieties is insufficient to meet the increasing demands of the growing population [5]. Thus, mutagenesis remains one of the most effective tools in functional genomics, facilitating the systematic identification of gene functions through two primary approaches: loss-of-function and gain-of-function mutagenesis [7,10,11,12].

This article examines the available mutant resources and highlights the potential of both loss-of-function and gain-of-function mutagenesis strategies in developing mutant libraries to advance rice functional genomics. Such mutagenesis approaches are essential for elucidating gene functions and enhancing critical traits in rice, a staple crop of considerable agricultural significance. The article also addresses the challenges associated with various mutagenesis techniques and underscores the importance of DNA barcoding for large-scale mutagenesis studies and OMICS technologies to accelerate rice functional genomics.

## 2. Approaches for Establishment of Loss-of-Function Mutants

### 2.1. Chemical and Physical Mutagenesis

To create genetic variability and identify the gene functions that cause changes in traits, random mutagenesis using chemicals, such as EMS, N-methyl-N-nitrosourea, sodium azide, and ENU, and physical agents, such as γ-rays, fast-neutron irradiation, ion beam radiation, and X-rays, are commonly employed to develop large-scale random mutations [13]. In chemical and physical mutagenesis, different plant parts, such as the embryo, seed, callus, and suspension culture, exhibit varying efficiencies, with seeds being the most used [10,12,13]. The dosage and duration of exposure to mutagens also significantly influence the efficiency of both chemical and physical mutagenesis [14,15,16,17].

Targeting induced local lesions in the genome (TILLING) reveals that 95% of chemical mutations are silent/missense, while physical mutagenesis often causes 15 bp deletions or chromosomal rearrangements [18]. For example, in fast neutron (FN)-mutagenized rice (Kitaake), there are 91,513 mutations affecting 32,307 genes, with 48% of these mutations having a single base change due to substitutions and deletions, while 66% of the mutations have other kinds of chromosomal rearrangements [19,20]. Moreover, chemical mutagens like N-methyl-N-nitrosourea (MNU) and ethyl methane sulfonate (EMS) are also essential tools for producing single-nucleotide polymorphisms with random distributions. In the indica rice variety IR64, for example, approximately 60,000 mutants induced via chemical and irradiation methods have facilitated significant advances in gene mining and functional characterization [21].

The genome-wide chips and high-throughput genotyping technologies indicate that there are larger chromosomal aberrations or deletions of essential genes in the mutant lines, limiting their effectiveness in rice mutagenomics [22]. While these methods facilitate large-scale mutagenesis, the identification of specific mutation sites and studies of gene functions across the genome through genetic mapping approaches can be laborious and time-consuming [23]. In contrast, T-DNA insertional mutagenesis offers a valuable alternative for generating loss-of-function mutants, as it tags mutations with identifiable inserted fragments, simplifying the process of mutation identification and functional analysis [24].

### 2.2. T-DNA Insertional Mutagenesis

In rice, the most widely used mutagenesis approach is insertional mutagenesis, where a segment of DNA is randomly inserted into the genome, leading to gene disruption and loss of function [25] (Figure 1). This DNA can be a transposable element (like Ac/Ds, Spm, and Tos17), which is a mobile genetic element capable of moving from one location to another within the genome when it has access to the necessary transposase enzyme [26,27], or it may consists of foreign DNA introduced into the cell [28]. This mechanism can lead to diverse genomic alterations, including deletions, insertions, and chromosomal translocations, thereby generating a wide range of mutant phenotypes [29]. Given the known sequence of the DNA insertional elements, the insertion sites in the mutants can be easily identified by using techniques like inverse PCR [30], thermal asymmetric interlaced PCR (TAIL PCR) [31], adapter ligation PCR [32], and DNA sequencing [30,33], which greatly facilitates the association of the phenotype with mutated genes. In 2017, researchers developed a barcode method for the pooled identification of transposon tagging sites using Illumina sequencing, and this can trace back the insertion sites to the preliminary individual clones of the transposon mutant library [34].

T-DNA insertional mutant libraries have been extensively constructed. These include 22,090 T-DNA insertion lines [28], 50,000 activation tagging lines of Nipponbare [35], 47,932 Kitaake activation tag lines [36], 171,000 T-DNA enhancer trap lines of different japonica backgrounds (Zhonghua 11, Zhonghua 15, Nipponbare) [37,38,39], and 22,665 Tainung 67 trifunctional T-DNA insertional lines [40]. The emphasis on japonica is due to its high transformation efficiency, enabling the large-scale generation of mutant resources [23,41]. Transposons have several advantages over T-DNA. In the Ac/Ds system, the Ds element preferentially transposes to genetically linked sites on the same chromosome, facilitating the establishment of an indexed insertion mutant library [42]. Tos17 is the most active endogenous retrotransposon for functional gene analyses [5]. In addition to T-DNA mutant lines, various transposon mutant resources have been established, including 50,000 mutant lines created through insertional mutagenesis [5]. A variety of genes have been identified through different mutagenesis approaches. For example, Tos17 helped to discover the photomorphogenesis regulator *phyA* [43]. Transposon-based mutagenesis, such as that with the Ac/Ds system, revealed key genes, including AID1 for another development trait [44], *OsCYP96B4* for cell elongation and lipid metabolism [45], *OsKS1* for germination and root growth [46], and *OsMYOXIB* for pollen development [47]. T-DNA insertion mutagenesis also uncovered important regulatory genes, such as *RID1* as a master regulator of flowering induction [48], *JMJ706* for floral development [49], and *OsCHLH*, which is involved in chlorophyll biosynthesis [50]. Other significant genes include *ILA1* for mechanical tissue formation [51], *DSM2* for the xanthophyll cycle and ABA synthesis [52], *OsPMT16* for pistil development [53], *MPK6* for early embryo development [54], *DTC1* for programmed cell death in the tapetum [55], and *OsPPDKB* associated with floury endosperm formation [56]. Overall, these mutagenesis approaches have greatly enhanced our understanding of the genes associated with important traits in rice, offering valuable insights for rice improvement.

In addition to the disruption of gene functions, the integration of DNA insertional mutagens was also modified to identify regulatory elements of genes via gene trapping, enhancer trapping, and promoter trapping technologies [23,57]. In the enhancer trap method [58], a reporter gene is placed near an enhancer to reveal gene expression patterns and regulatory networks. Promoter traps [23] involve introducing a reporter gene without its own promoter, but alongside a transcription unit, enabling it to be activated by adjacent promoters. This approach is valuable for functional analyses. Additionally, in gene trapping [28], a reporter gene is integrated into the genome, and its expression indicates that a gene is disrupted.

Currently, only 5% to 10% of the corresponding genes have been identified from the millions of mutant lines, as the integration of transposons or T-DNA does not always correlate with a change in the phenotype [59]. In addition to the demands of huge labor and the high costs of insertional mutagenesis manipulation, achieving whole-genome saturation via insertional mutagenesis suffers greatly from the unequal distribution of insertions caused by T-DNA and transposons throughout the genome [25,60,61]. For example, T-DNA prefers to integrate into the 5′ upstream and 3′ downstream regulatory regions and noncoding genes. Hence, inducing the coding region to have a single insertion tag is very challenging [25,60,62]. Furthermore, insertional mutagenesis is restricted to the targeting of multiple genes within the same pathways in a single mutant line. This constraint hampers the identification of functionally redundant genes [63].

### 2.3. Large-Scale CRISPR/Cas9-Mediated Mutagenesis

Several rice mutant resources have been established globally; however, traditional loss-of-function mutagenesis approaches face challenges, such as random integration and the multiple individual mutants needed to achieve mutations [63,64,65]. The *CRISPR/Cas9* genome editing system has emerged as a promising alternative, enabling targeted mutagenesis with low off-target effects [64]. CRISPR utilizes guide RNA (gRNA) to bind to the target DNA at the protospacer adjacent motif (PAM) and instructs Cas9 to perform specific cleavage [66,67] (Figure 2). The cell’s DNA repair mechanism then either uses nonhomologous end-joining (NHEJ) or homologous re-combination repair mechanisms [67], resulting in insertions and deletions (indels) that alter gene functions [68,69].

Initially, small-scale gene-knockout mutagenesis with CRISPR targets one or a few genes, which can be challenging due to the gene family [63]. However, *CRISPR/Cas9* can target many DNA fragments, enabling the rapid construction of large-scale knockout mutant libraries for screening. High-throughput loss-of-function screening using this system is quickly being applied to plants [64,70,71] (Table 1). By using multiplexing and pooled gRNA libraries, multiple redundant genes can be simultaneously mutated across the genome [64,72].

Recently, a unique tag approach has been developed for the identification of gRNAs, facilitating pooled CRISPR library construction and simultaneous transformation. This new approach successfully generated 955 *CRISPR/Cas9* mutant lines targeting the receptor like kinase (RLK)-family genes of rice, achieving a mutation frequency of 74.3% and enabling the identification of genes such as *OsRLCK109*, *OsCERK1*, *OsMRLK13*, *OsRLCK253*, *OsWAK32*, *LOC_Os01g03370*, *LOC_Os01g04580*, *LOC_Os03g56160*, *LOC_Os07g35300*, *LOC_Os09g18360*, and *LOC_Os09g37834*, involved in rice immunity [75]. Similarly, this approach has led to the development of 1319 *OsNAC*-gene-mutant rice lines (T0) with over 70% mutation frequency, aimed at uncovering the transcription factors (*OsNAC30*, *OsNAC59*, and *OsNAC101*) associated with rice immunity [76]. Large-scale *CRISPR/Cas9* loss-of-function mutagenesis efforts have increasingly relied on pooled gRNA libraries to identify various gene functions. For instance, a 88,541 gRNA library generated 84,384 ZH1 (*Oryza sativa* L. ssp. *Japonica*) mutant lines (T0) with a mutation frequency of 83.9% [72]. Additionally, the library of 25,604 gRNAs produced 14,000 Zhonghua 11 (*Oryza sativa* L. ssp. japonica) mutant lines (T0) with 86.4% targeted mutations [81]. Furthermore, 2184 large-scale *CRISPR/Cas9* mutant lines (T0) were developed by integrating 367 gRNAs targeting rice seed preferred genes, achieving a mutation frequency of 84.06% [82]. Despite these advancements, the number of mutant lines generated has remained low compared to other loss-of-function mutant libraries, and there is still a smaller number of mutant libraries. Therefore, the ongoing development of high-throughput, large-scale *CRISPR/Cas9* mutant lines is essential for identifying new gene functions, ultimately contributing to rice improvement.

### 2.4. Large-Scale RNA-Interference-Mediated Mutagenesis

The RNA interference (RNAi) technique is an essential mutagenesis tool in functional genomics that allows for the analysis of target gene effects through suppression [83] (Figure 2). It begins with double-stranded RNA (dsRNA) or hairpin RNA (hpRNA), which are processed by Dicer or Dicer-like (DCL) proteins into siRNAs or miRNA. These then associate with an Argonaute (AGO) protein, forming active effector complexes, miRNP or RISC, which repress expression of the corresponding target mRNA [84,85].

To establish genome-wide long hpRNA mutant libraries targeting multiple transcripts simultaneously, an approach called a rolling cycle amplification-mediated hpRNA (RMHR) system has been developed to target multiple genes of the plant. Using the RMHR approach, 1500 hpRNA rice mutant lines (T0) were developed, and they exhibited more than 47.9% of distinct phenotypic changes compared to the wild-type [41]. This indicates that mutation occurrence at the exonic sequence is much higher compared with the chemical and T-DNA insertional mutagenesis approaches [22,23,25,86]. Thus, this approach of mutant library construction serves as an effective tool for large-scale genome-wide gene function analyses (Table 1). While loss-of-function mutagenesis approaches are effective, two key challenges remain: (1) functional redundancy (approximately 29% of rice genes belong to multiple families, where the knockout of a single gene may not result in a clear phenotype due to functional complementation or buffering against the mutations [87,88,89,90]), and (2) essential genes for which the complete knockout causes lethality due to defective gametes, embryos, and growth and development [35,91,92]. RNA interference (RNAi) enables the study of essential genes at specific developmental stages without lethal effects [81,93]. For redundant gene families, pooled CRISPR/Cas9 or RNAi libraries can simultaneously target multiple members, overcoming compensatory effects [41,64,82,93,94,95,96].

## 3. Approaches for Establishment of Gain-of-Function Mutants

### 3.1. Activation Tagging

Activation tagging is one of the efficient functional acquisition mutations for the development of new phenotypes and to understand the genes that cause the change. In this system, T-DNA or a transposable element containing enhancers activates the expression of nearby genes upon insertion into the genome (Figure 3). The enhancer typically consists of tetrameric repeats of the CaMV35S promoter, which can activate gene expression in either orientation or extend up to 10 kb from the insertion site in the genome [35,90,97].

Activation-tagged mutagenesis approaches have been utilized in various species, including *Arabidopsis thaliana* [98], rice [99], wheat [100], oat [101], tomato [102], and barley [103], for systematic gene function identification.

In rice, significant mutant lines have been developed from different cultivars via T-DNA or Ac-Ds transposon mutagenesis. For example, the Dongjin and Hwayong cultivars with T-DNA yielded 47,932 lines [36,104], while Tainung 67 produced 55,000 lines [105,106]. The Nipponbare cultivar alone generated 13,000 lines with T-DNA [107] and 638 lines with the Ac-Ds system [108], plus an additional 50,000 lines from other T-DNA treatments [35]. These resources have advanced rice functional genomics despite challenges in development. Several genes have been identified for rice improvements through activation tagging [97], such as *GLR1*, *GLR2*, *XPB2*, *SEN1* for drought resistance [109,110], and *SPL18/OsAT1* linked to enhanced blast resistance [107]. Activation tagging has also revealed potential sheath blight (ShB)-resistance genes encoding glycoside hydrolase and Class III chitinase in indica rice [111]. However, the enhancer in the activation tagging can activate multiple genes up to 12.5 kb from the insertion site, complicating the connection between specific genes and their phenotype due to the simultaneous transcription of multiple endogenous genes [78,90].

### 3.2. FOX (Full-Length cDNA Overexpression) Hunting

FOX hunting is another valuable gain-of-function mutagenesis approach. In these approaches, full-length (Fl) cDNA libraries are cloned between the strong promoter and a terminator, followed by transformation, to generate the FOX lines [79,80,112]. In studies utilizing Arabidopsis, mutation rates for the FOX approach range from approximately 10% to 17% [79,80]. In contrast, the activation methods yield lower mutation rates, at around 1% [78]. Thus, the FOX lines serve as a vital resource for the identification of new genes, the characterization of proteins, and conducting biochemical analyses [112,113,114,115].

In 2006, Ichikawa et al. [79] developed 15,000 FOX Arabidopsis lines driven by the dicot-specific promoter CaMV35S, and the lines exhibit stable phenotype mutations, unlike the mutant lines produced by the loss-of-function mutation techniques [79]. Similarly, the rice FOX-Arabidopsis population, developed with 13,000 rice FL-cDNAs and the CaMV 35S promoter, generated an impressive 23,715 FOX lines, with 11,000 integrated FL-cDNAs, resulting in an integration efficiency of 84.6% [112]. To further elucidate rice gene functions, 13,980 rice FL-cDNAs were integrated into rice using the maize Ubiquitin-1 promoter, which resulted in 11,582 integrated lines and achieved an integration efficiency of 39.1% [80]. Additionally, 13,823 rice FL-cDNAs were driven by the rice Actin-1 promoter, yielding 1920 FL-cDNA-integrated lines with a 13.8% integration efficiency [116]. There were methodological differences in the binary vector, promoter, and cloning methods in these two cases. Despite the widespread use of in vitro site-specific Gateway recombination technology for establishing full-length cDNA libraries across various organisms, the integration rates of full-length cDNAs are lower compared to restriction site cloning techniques using the pRiceFOX binary vector [80,117,118]. Since its invention, the FOX hunting system has primarily focused on model plants, like Arabidopsis and rice. However, this technique is now applicable to a wide range of interspecific and intraspecific species, regardless of the genome size [112], including woody plants [119,120] and cereal crops [80]. This broader application enhances our understanding of gene functions and improves the genome annotation quality [115,121].

For example, approximately 15,000 FOX rice lines derived from 1455 full-length cDNAs of the transcription factors of the cultivated and the wild species of wheat have been established. Of these, 14.88% exhibited visible phenotype changes, with five lines demonstrating enhanced tolerance to salt and osmotic stress [122]. In the same way, Fujita et al. [123] established an Arabidopsis FOX line using a mix of 43 salt-induced full-length cDNAs of transcription factors, leading to identification of the *bZIP* transcription factor-related gene *AtZIP60*, which plays a role in regulating the salt response. This implies that the FOX hunting approaches can effectively facilitate gene function identification within and across both wild and cultivated species. However, like other insertional mutagenesis techniques, this approach also requires high-throughput screening of the full-length cDNA, as well as high-throughput phenotyping and genotyping of mutant lines, to advance rice functional genomics (Table 1). The costs and the technical challenges associated with the existing full-length cDNA library synthesis for FOX hunting are also challenging.

## 4. Genetic Resources Required for a Gain of Function

### 4.1. Full-Length cDNA Library

So far, more than 60,000 full-length cDNAs, including about 38,000 full-length cDNAs from japonica cv. Nipponbare [124], 10,096 FL-cDNA sequences from indica variety (Guangluai 4) [125], and 10,828 from indica variety (Minghui 63) [126], and with only around 1888 putative full-length cDNA clones, have been completely sequenced from wild rice (*Oryza rufipogon Griff. W1943*) [127]. However, many existing FL-cDNAs lack detailed functional annotations and are insufficient for the functional characterization of all rice genes. Despite the efforts of full-length cDNA synthesis, it is still hindered by technical issues, such as incomplete transcripts, sequencing costs and errors, and difficulties in capturing alternative splicing events. Thus, these challenges require new approaches to cover and efficiently translate genomic data into functional insights.

In establishing large Fl-cDNA libraries in rice, different approaches yield varying success rates for full-length clones. For instance, oligo capping methods achieve over 90% full-length cDNA clones [124], while cap tagging yields approximately 70% [127]. In contrast, conventional oligo dT or adapter-oriented methods result in only 60% full-length cDNA clones [126]. The choice of the techniques is vital, as it directly impacts the quality and completeness of the resulting cDNA synthesis.

Alternatively, genomic DNA can be used directly as a template for single exon-containing genes, to enrich the amount of rice Fl-cDNA libraries, and offer significant promise for functional genomic studies and explorations of gene functions. This approach simplifies the process because the entire gene is contained within a single exon, eliminating the need for complex splicing or the handling of introns. The benefits of this method include reduced complexity and cost-effectiveness, as it minimizes the steps involved in isolating mRNA and converting it to cDNA. In the rice genome, more than 11,109 genes (19.9%) are monomorphic (single exon genes) and belong to a large gene family, like F-box proteins, ribosomal proteins, DEAD-box helicase, and so on [128]. The single-exon genes are much more beneficial for regulatory activities since they have better transcriptional fidelity due to the absence of splicing events [129,130].

### 4.2. Wild Relatives of Rice

It is noteworthy that there are currently very few, if any, libraries of rice lines available that are derived from wild species, which limits our ability to investigate novel gene functions from rice ancestors. Gain-of-function mutagenesis is a transformative approach to crop improvement that leverages the wild relatives’ genetic diversity and enhances specific traits in cultivated varieties [114,122,131]. Nowadays, climate variability and biotic and abiotic stress threaten rice production globally. Thus, the genetic variability of Oryza wild relatives will provide game-changing traits and genes to address the challenges of resistance and global agricultural production in current and future climates [132,133].

For an understanding of the function of wild rice, Lu et al. [127] has established 1,888 full-length cDNAs from wild rice (*Oryza rufipogon Griff. W1943*), and the homology sequence analysis showed that more than 96.8% of the wild rice full-length cDNA is highly homologous with the cultivated rice genome sequence and that 1% (17) of the full-length cDNA comprises unique rice genes that can be used for further study to advance rice functional genomics and breeding. Very recently, the functional screening of over 4000 wild apple (*Malus sieversii*) FOX Arabidopsis lines identified a total of 160 inserts appearing to be novel, with no or limited homology to the cultivated apple (*Malus pumila*) and Arabidopsis and mainly providing freezing resistance [134]. This implies that the wild genetic resource can also be used as a genetic donor for gain-of-function mutagenesis approaches for new gene discovery.

## 5. DNA Barcoding for Large-Scale Mutagenesis to Accelerate Functional Genomics

### 5.1. Fast and High-Throughput Genotyping

To utilize the mutants generated, genotyping the mutants is essential. Traditional technologies largely rely on PCR amplification and Sanger sequencing, which make the mutant genotyping time-consuming and tedious. Recently, DNA barcoding has emerged as a revolutionary method that enables the simultaneous sequencing of pooled samples from numerous genetic materials (Figure 4). These fixed-length-index nucleotide sequences allow for efficient multiplexing during sequencing [135], making it crucial for biodiversity studies across various taxa [136,137], detecting invasive species [138], understanding plant–pollinator interactions [139,140], and assessing water quality [137].

The future of functional genomics using barcoding techniques is promising, as unique sequences help to avoid redundancy and simplify identification [2,141]. Sample identifiers can be added via ligation or with a one-step PCR process [136,141,142], two-step PCR, offering flexibility with potential contamination risks [136,141,142], and tagged PCR approaches, which provides more design flexibility [136,143] while minimizing biases, although it risks tag-jumping errors [136].

Barcode libraries enhance functional genomics by allowing for the high-throughput screening of gene functions (Table 2). They consist of unique DNA sequences that enable the tracking of individual genes or genetic elements. Combining forward and reverse genetics with DNA barcoding is vital for understanding gene functions and interactions, especially in large-scale libraries for biomolecule identification [142]. Loss-of-function CRISPR/Cas9 mutant lines are significant for gene function discovery. High-throughput screening using sgRNA with unique identifiers has improved the efficiency in pooled mutant screening [144,145]. The barcode-based NGS approach successfully tracked mutations in maize, revealing genes with observable phenotype changes [146]. Additionally, a barcode-tagged insertion mutant library in yeast has enabled the high-throughput genotype analysis of 10,000 mutants [147].

### 5.2. High-Througput Phenotyping and OMICS Integration to Accelerate Functional Genomics

Many of the mutant lines are complemented by phenotypic information. The association of genetic and phenotypic data has propelled significant progress in all-inclusive analysis and enhancing our understanding of gene functions [24]. However, the phenotyping of mutants still depends on the manual scoring of visible traits. Thus, the ongoing large-scale developments of rice mutant resources have highlighted the necessity of high-throughput phenotyping to tackle the challenges of identifying the mutant phenotypes developed through loss- and gain-of-function mutations [40]. High-throughput phenotyping (phenomics) is being used for multidimensional phenotypes from cellular-to-organ-level assessments and can be extended to individual plants and entire fields for the identification of gene functions and new genes/QTLs and the understanding of the genetic architecture of a complex trait and environmental responses [154,155,156,157]. This approach can also increase the efficiency, reduce the labor, and analyze large populations to make associations between the genotype and phenotype [158].

Recently, the rapid advancement of the use of image-based approaches, implementation of robotic and conveyer belt and ground based, and aerial imaging have enhanced the invoice monitoring of plants [155]. For instance, Yang et al. [159] established a high-throughput nondestructive automatic phenotyping imaging device capable of monitoring fifteen agronomic traits, including plant compactness and grain projection areas of 5472 rice growing pots, leading to the identification of green revolution semi dwarf (SD1) genes through genome-wide association studies (GWASs). Additionally, high-throughput phenotyping relies heavily on imaging technologies, including multi-angle imaging for rice panicle numbers [154], structured light 3D imaging for filled and unfilled grain identification [160], and light emitting diode (LED) transmission imaging for rice yield trait characterization [161]. Additionally, 51 image-based traits related to drought responses were analyzed, using nondestructive high-throughput phenotyping and the identification of a new drought resistance gene (*OsPP15*) through genetic transformation experiments [162].

The integration of phenotype and OMICS technology improves the mutant characterization and identification of gene functions. For instance, the elemental profiling (Ionomics) of rice FOX Arabidopsis mutant lines revealed that *OSZIP* increases the Zn concentration [163], while metabolomic screening revealed that *Os-LBD37/ASl39* is involved in nitrogen metabolic processes [113]. Additionally, the elemental profiling of T-DNA insertion mutant lines showed that *OsNAC5* lowers iron and zinc concentrations in the leaves while increasing iron levels in the seeds compared to the wild-type [164]. Transcriptomic analyses revealed that the *OsNRAMP5* gene is responsible for low Cd uptake and accumulation in EMS-mutagenized indica rice (9311) [165] and ion-beam irradiated seeds of japonica rice (Koshihikari) [166]. Furthermore, the ability to identify gene interactions further enhances the effectiveness of mutagenesis approaches. For example, loss-of-function studies, such as knocking out the *OsMADS57* gene in an antisense manner, can demonstrate decreased tiller numbers in rice. However, proteomic and interatomic analyses reveal that complex interactions with OSTB1 proteins modulate rice tillering through DWARF14 [167]. The absence of such OMICS integration can challenge the understanding of compensatory mechanisms that may arise from the knockout.

Indeed, while gain-of-function and loss-of-function mutagenesis methods are valuable tools in rice functional genomics, their effectiveness for the advancement of rice functional genomics can substantially advance with the integration of high-throughput phenotyping, genotyping, and OMICS technologies.

## 6. Conclusions and Future Directions

The systematic identification of gene functions critically depends on phenotypic observations of loss-of-function or gain-of-function mutants. As demonstrated throughout this review, the establishment of large-scale mutant libraries has enabled genome-wide functional annotations at unprecedented scales. These resources have proven to be particularly valuable for overcoming the limitations of traditional single-gene approaches, allowing us to address complex biological questions regarding gene redundancy (e.g., via multiplexed CRISPR/Cas9 libraries), essential gene functions (e.g., via RNA interference and FOX hunting), and precise gene–trait relationships. Approximately 30% of rice genes belong to functionally overlapping families, obscuring the phenotypic effects in single-gene knockouts. The integration of high-throughput technologies has enhanced functional genomics. DNA barcoding coupled with next-generation sequencing has improved the efficiency of genotype–phenotype linkages, while high-throughput phenotyping helps for large-scale gene discovery.

Future research should prioritize the following: (1) development of scalable mutagenesis approaches. Multiplexed CRISPR/Cas9 mutagenesis enables the targeting of redundant gene families, while prime and base editing technologies can generate allelic series to study essential genes without lethal effects, overcoming the lethality hurdle of knockouts. These approaches will provide the necessary tools to achieve complete functional annotations of the rice genome; (2) conditional pooled RNAi knockdown to study lethal mutations. This approach complements CRISPR technologies by allowing for the temporal and spatial control of gene suppression, which is particularly valuable for investigating essential biological processes; (3) optimization of the DNA barcoding kit. This toolkit will be essential for large-scale library construction and mutant tracking. The use of multiplexed high-throughput sequencing will further enhance the capacity to manage and analyze ever expanding mutant collections for accelerating the rate of gene discovery; and (4) expanding the FOX libraries with wild rice (e.g., *Oryza Rufipogon*) cDNA to unlock novel alleles/genes. The utilization of wild genetic resources during mutant library establishment will help to discover potentially valuable alleles that have been lost during domestication processes. This approach may offer critical traits for tolerance and yield improvements. The integration of scalable mutagenesis approaches with high-throughput phenotyping (e.g., automated imaging) and multi-OMICS (e.g., metabolomic, ionomics) approaches will enable large-scale mutant characterization and accelerate the genomic discoveries and breeding applications.

## Figures and Tables

**Figure 1 plants-14-01492-f001:**
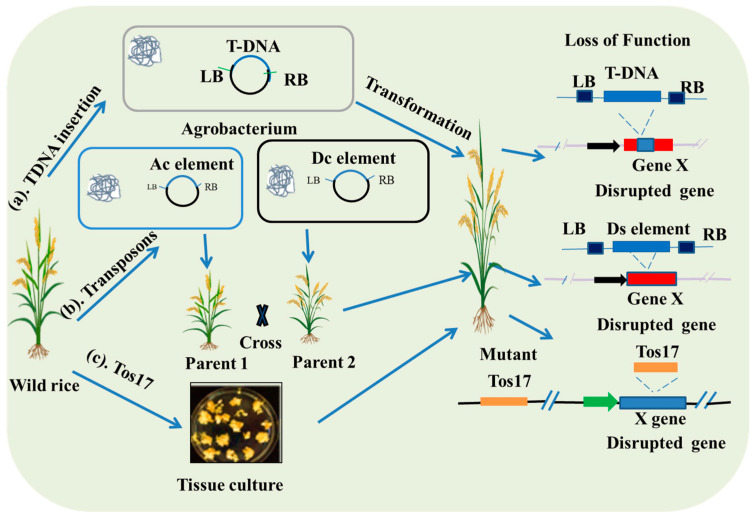
Insertional mutagenesis approaches to develop loss-of-function mutants. This figure illustrates three distinct approaches for developing loss-of-function mutants. (**a**) The T-DNA insertion method utilizes *Agrobacterium tumefaciens* harboring the binary vector to genetically transform rice, leading to the integration of T-DNA into the genome and the disruption of gene functions. (**b**) The Ac/Ds mutant lines are generated by crossing independently transformed Ac and Ds lines, which facilitates the disruption of gene functions in the subsequent generations. (**c**) The Tos17 insertion method involves tissue or callus culture activation to integrate Tos17 into the genome, resulting in the disruption of gene functions.

**Figure 2 plants-14-01492-f002:**
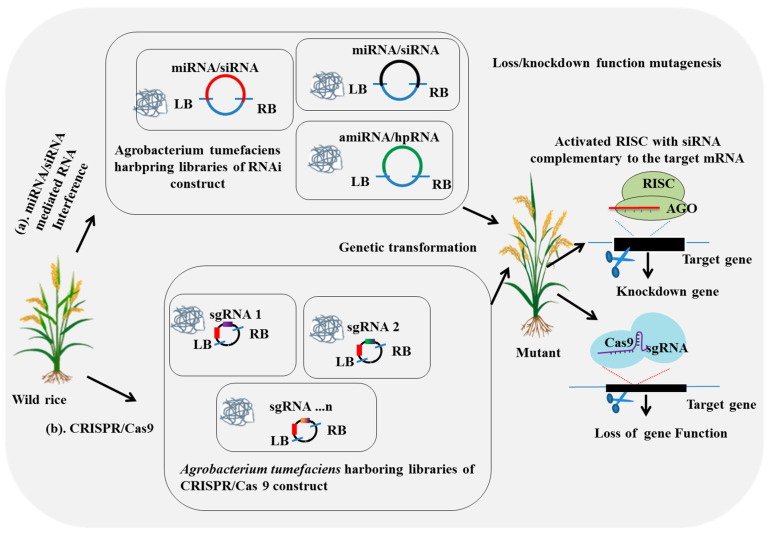
miRNA/siRNA-mediated RNA interference and CRISPR/Cas9-mediated mutagenesis approaches. This figure describes two advanced approaches for knockdown and loss-of-function mutagenesis. (**a**) The miRNA/siRNA mediated RNA silencing method involves genetic transformation using Agrobacterium that harbors pools of miRNA/siRNA constructs, enabling the generation of large-scale knockdown-mutant lines. (**b**) The CRISPR/Cas9-mediated mutagenesis approach utilizes genetic transformation with Agrobacterium containing pools of sgRNA constructs, facilitating the creation of large-scale knockout-mutant lines.

**Figure 3 plants-14-01492-f003:**
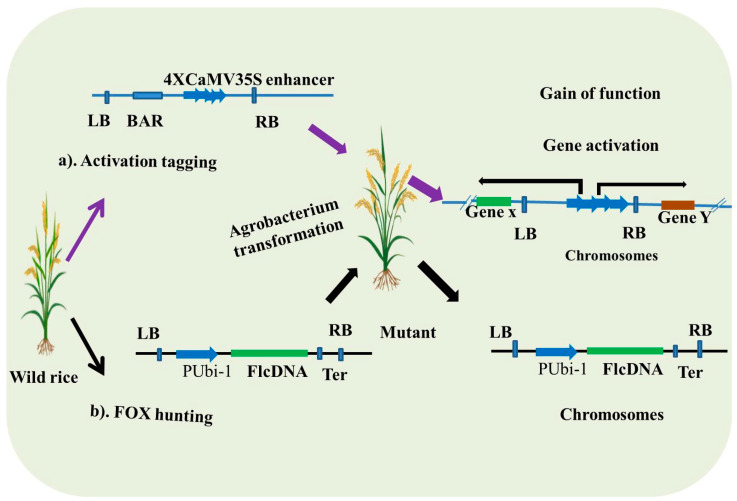
Gain-of-function mutagenesis approaches. This figure illustrates two key gain-of-function mutagenesis approaches. (**a**) Activation tagging: this approach involves the insertion of a strong enhancer construct, using *Agrobacterium tumefaciens* transformation, near a gene of interest, thereby enhancing its expression and resulting in the gain of function. BAR, glufosinate resistance; LB, left border; RB, right border. (**b**) FOX hunting. In this approach, full-length cDNA is cloned between a strong promoter and terminator and then transformed using *Agrobacterium tumefaciens* to drive the overexpression of specific genes. PUbi-1, ubiquitin promoter; Ter, terminator.

**Figure 4 plants-14-01492-f004:**
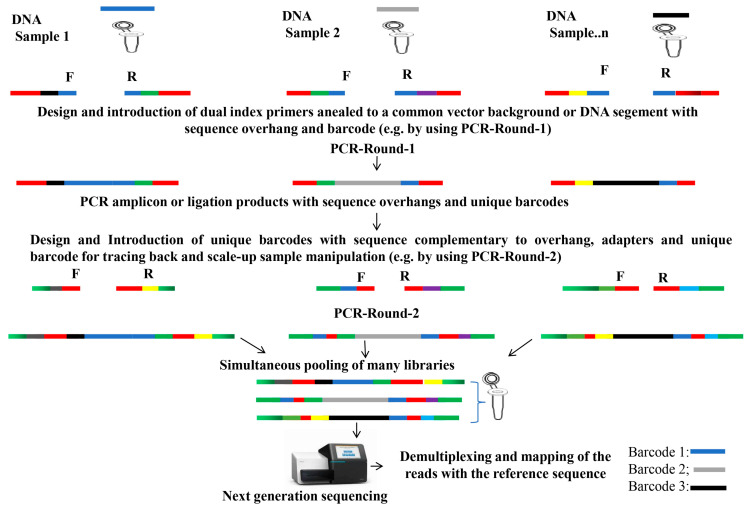
This figure illustrates a simplified dual-index barcoding approach, facilitating the simultaneous pooling of various sample types, including cDNA libraries for FOX hunting mutagenesis, gRNA for CRISPR/Cas9-mediated mutagenesis, and genomic DNA for identifying transposon or T-DNA insertion sites. This innovative design enhances the efficiency of next-generation sequencing (NGS) by allowing for the multiplexing of samples, thereby reducing costs and improving the throughput in genotyping. The color is to indicate the unique combinatorial barcoding method.

**Table 1 plants-14-01492-t001:** Comparative analysis of mutagenesis approaches for large-scale mutant analysis.

Mutagenesis Tools	Mutation Type	Advantages	Challenges	Precision	Scalability	References
Chemical (e.g., EMS)	Random	Simple non-transgenic with high mutagenesis efficiency	Requires extensive screening	Low	Low	[73]
T-DNA Insertion	Random	Genome-wide coverage; barcode-compatible. Thousands of mutants analyzed in parallel	Random insertion disrupts non-target genes.Individual mutant screening via PCR or phenotyping	Low	High	[24,28]
Transposons (e.g., Ac/Ds)	Random	Mobility enables regional mutagenesis.Reusable systems	Requires transposase control; lower efficiency Individual mutant screening via PCR and phenotyping	Moderate	Moderate	[74]
CRISPR/Cas9	Targeted knockouts/editing	High precision; multiplex editing of redundant genes	Complex design for large libraries	High	High	[64,75,76]
RNAi	Targeted knockdown	Rapid; scalable with barcoded vectors (e.g., pooled RNAi screens) for redundant genes	Requires efficient design for target silencing	High	Moderate	[41,64,77]
Activation tagging	Gain-of-function	Identifies dominant alleles	random activation may cause pleiotropy Low mutant frequency	Low	Moderate	[35,78]
FOX hunting	cDNA overexpression	Precise overexpression. Possible to use barcoding to trace back the cDNA clones; links phenotype to known genes	High cost of cDNA library construction. Complex phenotyping	High	High	[79,80]

**Table 2 plants-14-01492-t002:** Important features of DNA barcode-based large-scale mutagenesis approaches to accelerate functional genomics.

Approaches for Integration of DNA Barcoding		References
FOX hunting	Pooled cDNA clone libraries or mutants with unique barcodes can be synthesized using amplification primers or sequencing primers	[148]
Activation tagging	Integrates unique barcodes into the T-DNA construct	[24]
T-DNA	Embeds unique barcodes into the mutagenic T-DNA construct	[24]
Transposon (Ac/Ds)	Ds transposons tagged with barcodes for tracking reinsertion sites	[24,149]
Retrotransposon (Tos17)	Unique barcode embedded with long terminal repeat (LTR) for tracking insertion sites	[150]
RNAi	Multiplexed shRNA or amiRNA libraries with unique barcodes	[41,77]
CRISPR/Cas9	Multiplexed sgRNA libraries with unique barcodes	[75,76,82]
Important features of DNA barcoding		
Tracking and screening	Scalable: sequencing identifies barcodes linked to samples	[141,151]
Scalability	Thousands of mutants can be analyzed simultaneously	[24,41,70,75,76,77,82]
Precision	High: pooled libraries enable genome-wide screening (e.g., CRISPR-Cas9 with barcoded sgRNAs; and shRNA or amiRNA libraries with unique barcodes)	[41,75,76,77,144]
Applications	Large-scale functional genomics, pooled screens	[24,41,70,75,76,77,82]
Challenges	Requires careful design and integration of DNA barcode and next-generation sequencing data analysis	[136,152,153]

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
