# Peer review of "Large-Scale Rice Mutant Establishment and High-Throughput Mutant Manipulation Help Advance Rice Functional Genomics"

_plants, 2025, doi:10.3390/plants14101492_

Round 1

Reviewer 1 Report

Comments and Suggestions for Authors

This manuscript reviews different mutagenesis in rice and aims to develop a mutant library for rice. Individual sections are written well, but the overall organization should be improved.  It is not clear what authors want to review 1) methods for mutagenesis, 2) functinoal characterization, or 3) building a mutagenesis library.

  1. Lines 21-23: Low mutation frequency probably helps to characterize genes. “Low mutation frequency” should be removed.
  2. The overall theme is not very clear. The title suggests building a mutant library is the key, but a large fraction of the main body is about different mutagenesis methods in rice. Or authors seem to focus on the functional annotation.  Authors have to make the overall flow consistent, including the title, abstract, and the main body.
  3. While authors emphasize that mutagenesis studies can elucidate the gain/loss-of-function mutations, it is difficult to do such analysis if a gene in not functionally annotated. It would be great if authors can show is the mutagenesis data can be used toward functional characterization of potential genes.  For example, random insertion data can suggest the location of insertion-protected regions, which is often a gene.    
  4. Information on Table 2 seems generic beyond the first group of approaches for barcoding. This feels like two tables are combined into one.
  5. Please, add the concluding remarks according to authors’ stance.

Author Response

  1. Comments 1: Lines 21-23: "Low mutation frequency probably helps to characterize genes. 'Low mutation frequency' should be removed."
  • Response: To make it clear, we have restructured the abstract. The change is highlighted in red in revised manuscript.
  1. Comments 2: The overall theme is not very clear. The title suggests building a mutant library is the key, but a large fraction of the main body is about different mutagenesis methods in rice. Or authors seem to focus on the functional annotation. Authors have to make the overall flow consistent, including the title, abstract, and the main body
  • Response: The systematic identification of gene function relies on large scale mutant libraries, which enable high through put phenotypic screening. This review (1) details mutagenesis approaches for constructing these libraries, (2) demonstrate the available mutant resources and construction challenges, and (3) indicated how integrating DNA barcoding and highthroughput phenotyping enables scalable functional genomics. We have modified the title and the abstract to better reflect this focus.
  • Original Title: Large-Scale Rice Mutant Library Construction for the Advancement of Rice Functional Genomics
  • Revised: Large Scale Rice Mutant Establishments and Highthrougput Mutants Manipulation help Advance Rice Functional Genomics
  1. Comments 3: While authors emphasize that mutagenesis studies can elucidate the gain/loss-of-function mutations, it is difficult to do such analysis if a gene is not functionally annotated. It would be great if authors can show is the mutagenesis data can be used toward functional characterization of potential genes. For example, random insertion data can suggest the location of insertion-protected regions, which is often a gene.
  • Response: We appreciate the reviewer’s insightful observation regarding the challenges of characterizing unannotated genes. While traditional small-scale mutagenesis typically focuses on annotated genes, our review discusses how large-scale approaches can systematically characterize both annotated and unannotated genes. Example, as highlighted in section 3.2, FOX hunting approach enables gene discovery by overexpressing full length cDNA without requiring prior annotations and identify gene function through phenotypic observation from uncharacterized cDNA clones. The integration of DNA barcoding (Section 5.1) also enables pooled screening of uncharacterized mutants, establishes genotype-phenotype links for unknown genes through high throughput sequencing and facilitates genome scale functional annotations.
  1. Comments 4: Table 2 seems generic beyond the first group of barcoding approaches. It feels like two tables combined.
  • Response: We restructured the Table 2 for more clarity. It is highlighted in red in Table 2 of the manuscript.

Reviewer 2 Report

Comments and Suggestions for Authors

The review is valuable and organizes knowledge on how to obtain plant mutants with emphasis on their performance. However:
- it is a bit too winded in some places,
- some figures and Table 1 need to be corrected or re-edited,
- several cited articles are not listed in the references,
- my other comments are highlighted in the attached file. 

Author Response

  1. Comments 1: it is a bit too winded in some places.
  • Response: Thank you for your feedbacks and constructive comments. In addition, with accepting the comments highlighted in the manuscripts, we also make some slight modifications.
    • Original (Line 39-45): However, the traditional breeding methods often face significant limitations in addressing specific challenges such as yield improvement, biotic and abiotic stress resistance, and nutritional enhancements.
    • Revised (line 44-47): Traditional breeding struggles to address yield, stress tolerance, and nutritional traits, necessitating advanced mutagenesis. The annual rice production growth has slowed to the point that it is no longer able to keep up with the growing global population of consumer [5].
    • Original (line 81-84): The analysis of targeting induced local lesions in genome (TILLING) for screening of mutations at the molecular level indicated that most of the chemically induced DNA lesions resulted silent and missense mutations (95%) and the physical mutagenesis treatments result in around 15bp nucleotide deletion, single base substitution, inversion and translocation
    • Revised (line 81-82): Targeting induced local lesions in genome (TILLING) reveals that 95% of chemical mutations are silent/missense, while physical mutagenesis often causes 15-bp deletions or chromosomal rearrangements
    • Original (line 232-256): However, unlike the CRISPR/Cas 9 approach, this method does not generate complete knockout mutants; it is primarily useful for studying gene function by reducing gene expression[74, 80]. Complete knock-out of essential genes is lethal to plants, making it impossible to recover such mutants using CRISPR/Cas 9 like mutagenesis approaches [77]. However, mutants of essential genes can be recovered for gene function analysis by incomplete gene knockdown with RNA silencing system[24, 81]. Overall, traditional methods of constructing loss of function mutant libraries have been proven to be effective but still face challenges for efficiently targeting genes with functional redundancy and those that are essential for growth and development[67, 82]. For instance, approximately 29% of the predicted 37,544 genes in rice are clustered and exhibit redundancy[83]. This redundancy complicates the observation of visible phenotypes and the study of the functions of these duplicated genes, as the loss-of-function mutations in some members of a multicopy gene family can be compensated by the remaining members through functional complementation or buffering against the mutations[19, 84, 85]. Additionally, many loss-of-function mutagenesis strategies face difficulties in producing mutant phenotypes, particularly for genes involved in multiple life cycle stages. Mutations in such genes often lead to early lethality due to defective gametes, embryos, and overall growth and development [34, 82, 86]. This early lethality presents a significant barrier to functional analysis as it restricts the ability to study the gene’s role throughout the plant life cycle. To overcome, these challenges conditional mutagenesis like RNA interference, can allow studying the essential genes at specific developmental stages, thereby circumventing early lethality[74, 80]. Additionally, the advancements of using large scale CRISPR/Cas 9 and hpRNA or miRNA mediated RNA interferences mutagenesis can create targeted mutations for functional study of redundant genes redundancy[40, 63, 75, 80, 81, 87, 88] .
    • Revised (line 231-240): While loss of function mutagenesis approaches are effective, two key challenges remain: (1) functional redundancy (approximately 29% of rice gene belong to multiple families where knockout of single gene may not result clear phenotype due to functional complementation or buffering against the mutations [81-84], and (2) essential genes whose complete knockout causes lethality due to defective gametes, embryos, and growth and development [35, 85, 86]. RNA interference (RNAi) enables to studying essential genes at specific developmental stages without lethal effects [75, 87]. For redundant gene families, pooled CRISPR/Cas9 or RNAi libraries can simultaneously target multiple members, overcoming compensatory effects [41, 64, 76, 87-90].
  1. Comments 2: Figures and Table 1 need corrections.
  • Response: Accepted the comments and edited the figures and the tables.
  1. Comments: several cited articles are not listed in the references.
  • Response: Yes, we agree. There were errors during formatting of the references for journal submission. Thus, we have fixed the citations and also added missing references.
  1. Comments: Other minor edits (highlighted in the attached file).
  • Response: Accepted and addressed all tracked changes with updated citations.

Round 2

Reviewer 1 Report

Comments and Suggestions for Authors

My previous comments were addressed well, and I support the publication as it is.

Author Response

We sincerely appreciate the time and effort you have dedicated to reviewing our manuscript. Your constructive comments have improved the clarity, and scientific precision of our work. Below, we provide a point-by-point response to the  comments of reviewer2'in PDF, to address their concerns and improve the manuscript.
